# Structural Differences of PM$_{2.5}$ Spatial Correlation Networks in Ten Metropolitan Areas of China

Shuaiqian Zhang [1], Fei Tao [1,2], Qi Wu [1], Qile Han [1], Yu Wang [1] and Tong Zhou [1,*]

1    School of Geographical Sciences, Nantong University, Nantong 226007, China;
     1921110012@stmail.ntu.edu.cn (S.Z.); taofei@ntu.edu.cn (F.T.); 2021110027@stmail.ntu.edu.cn (Q.W.);
     2021110033@stmail.ntu.edu.cn (Q.H.); 2021110041@stmail.ntu.edu.cn (Y.W.)
2    Department of Land Surveying and Geo-Informatics, The Hong Kong Polytechnic University,
     Hong Kong 999077, China
*    Correspondence: zhoutong@ntu.edu.cn; Tel.: +86-135-8521-7135

**Abstract:** The cross-impact of environmental pollution among cities has been reported in more research works recently. To implement the coordinated control of environmental pollution, it is necessary to explore the structural characteristics and influencing factors of the PM$_{2.5}$ spatial correlation network from the perspective of the metropolitan area. This paper utilized the gravity model to construct the PM$_{2.5}$ spatial correlation network of ten metropolitan areas in China from 2019 to 2020. After analyzing the overall characteristics and node characteristics of each spatial correlation network based on the social network analysis (SNA) method, the quadratic assignment procedure (QAP) regression analysis method was used to explore the influence mechanism of each driving factor. Patent granted differences, as a new indicator, were also considered during the above. The results showed that: (1) In the overall network characteristics, the network density of Chengdu and the other three metropolitan areas displayed a downward trend in two years, and the network density of Wuhan and Chengdu was the lowest. The network density and network grade of Hangzhou and the other four metropolitan areas were high and stable, and the network structure of each metropolitan area was unstable. (2) From the perspective of the node characteristics, the PM$_{2.5}$ spatial correlation network all performed trends of centralization and marginalization. Beijing-Tianjin-Hebei and South Central Liaoning were "multi-core" metropolitan areas, and the other eight were "single-core" metropolitan areas. (3) The analysis results of QAP regression illustrated that the top three influencing factors of the six metropolitan areas were geographical locational relationship, the secondary industrial proportion differences, respectively, and patent granted differences, and the other metropolitan areas had no dominant influencing factors.

**Keywords:** PM$_{2.5}$; social network analysis; metropolitan area; spatial correlation network

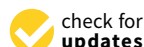



## 1. Introduction

With the advancement of urbanization, air pollution is not only a problem in China but also a serious problem in large cities, especially in the global south [1–3]. It harms human health and affects the development of the economy, ecology, and industry [4,5]. PM$_{2.5}$ (particles with diameter less than or equal to 2.5 microns), as the main pollutant affecting air quality, has become a research hotspot of scholars [6]. Neighboring cities have both competition and cooperation relationships. There is cross-impact of environmental pollution along with regional economic development. PM$_{2.5}$ pollution has an obvious spatial effect [7]. Therefore, it is significant in clarifying the PM$_{2.5}$ spatial correlation between cities for coordinated control of environmental pollution.

From the spatial scale, PM$_{2.5}$ pollution characteristics were initially studied in a single city, such as Beijing [8], Chongqing [9], Tianjin [10,11], etc. However, due to the complex formation and spatial heterogeneity of PM$_{2.5}$ concentration, the spatial distribution pattern of

PM$_{2.5}$ is usually different. Research on PM$_{2.5}$ in a single city ignored the spatial cross-impact characteristics of pollution [12,13]. Therefore, some scholars started from the perspective of a single urban agglomeration [14,15] or multiple urban agglomerations [16]. For example, Chen et al. [17] used PM$_{2.5}$ data and population data to analyze the spatiotemporal evolution of PM$_{2.5}$ concentration and population exposure risk in the Beijing-Tianjin-Hebei urban agglomeration. Zhu et al. [18] explored the impact of urbanization in the Yangtze River Economic Belt on PM$_{2.5}$. It was concluded that PM$_{2.5}$ in the Yangtze River Economic Belt had obvious spatial autocorrelation. Compared with urban agglomerations, the metropolitan area takes the central city with a developed economy and strong urban functions as the core. It is composed of the central city and the areas covered by several neighboring cities that have economic links with it [19]. The metropolitan area, as a region with more network characteristics than urban agglomeration, had more research significance. Therefore, the metropolitan area was determined as the research scale of this paper.

The spatial autocorrelation analysis [20,21] and standard deviation ellipse [22] have become common methods to explore the spatial distribution of PM$_{2.5}$. The land use regression (LUR) model [23], geographically weighted regression (GWR) analysis [24,25], and geographic detector [26] are mainly used to explore the influencing factors on PM$_{2.5}$ concentration. However, traditional econometric models are difficult for exploring the PM$_{2.5}$ spatial connection between cities and realizing the coordinated control of PM$_{2.5}$. The spatial correlation of PM$_{2.5}$ between regions in China is complex and has obvious network structure characteristics [27]. Therefore, this paper adopts the social network analysis (SNA) method to explore PM$_{2.5}$ spatial correlation, providing the theoretical basis for the joint control policy of PM$_{2.5}$ pollution.

A single influencing factor, such as population [28,29], urbanization level [30,31], and vegetation [32–34], was used to explore PM$_{2.5}$ pollution in a sample area in the early stage. As time goes on, some studies began to consider the coordinated effects of socioeconomic, meteorological, and other factors [35–39]. For example, Xu et al. [40] studied the distribution of PM$_{2.5}$ in the Yangtze River Delta by combining socioeconomic factors such as gross domestic product (GDP) and population density with three meteorological factors such as wind speed, precipitation, and temperature. However, the influencing factor system in the above studies is lacking, and the difference in scientific and technological innovation ability between cities will also contribute the driving factors system. Central cities with strong scientific and technological innovation ability tend to attract edge cities to carry out cooperation with them, thus affecting the correlation of PM$_{2.5}$. Therefore, the patent granted differences are taken as an influencing factor in this paper to represent the difference in the scientific and technological innovation ability of cities.

To sum up, this paper took ten metropolitan areas in China as the study area, chose 2019–2020 as the time range from an outbreak to the control of the novel coronavirus disease (COVID-19) in China, used PM$_{2.5}$ data under the National Monitoring System, combined meteorological data and socioeconomic data, and adopted the social network analysis method to study the structural characteristics and influencing factors of the PM$_{2.5}$ spatial correlation network. The research highlights were as follows: (1) It was the first time to reveal the spatial characteristics and differences of PM$_{2.5}$ from the perspective of two northern metropolitan areas and eight southern metropolitan areas in China. (2) Based on the PM$_{2.5}$ spatial correlation network, the social network analysis method was used to compare the overall network characteristics and node characteristics of each metropolitan area. Ten metropolitan areas were divided into "single-core" and "multi-core" metropolitan areas, and pollution control policies were put forward according to the actual situation of the city. (3) The patent granted differences, as a factor of scientific and technological innovation, was put into the quadratic assignment procedure (QAP) regression analysis method. The influencing factor system composed of the geographical locational relationship, the population density differences, the secondary industrial proportion differences, the tertiary industrial proportion differences, and mean annual maximum temperature differences were improved. According to the analysis results of influencing factors, the ten metropoli-

tan areas were divided into three categories: "mature stage of economic development", "growth stage of economic development" and "reform stage of economic development".

The rest of this article is organized as follows: In Section 2, the study area and data sources are briefly introduced, and the research method of this study is described. The experimental results are presented in Section 3, and Sections 4 and 5 are discussion and conclusions.

## 2. Materials and Methods

### 2.1. Study Area

This paper selects ten metropolitan areas in China, which are the South Central Liaoning metropolitan area, Beijing-Tianjin-Hebei metropolitan area, Nanjing metropolitan area, Suzhou-Wuxi-Changzhou metropolitan area, Hangzhou metropolitan area, Wuhan metropolitan area, Changsha-Zhuzhou-Xiangtan metropolitan area, Chengdu metropolitan area, Guangzhou-Foshan-Zhaoqing metropolitan area, and Shenzhen-Dongguan-Huizhou metropolitan area (Figure 1).

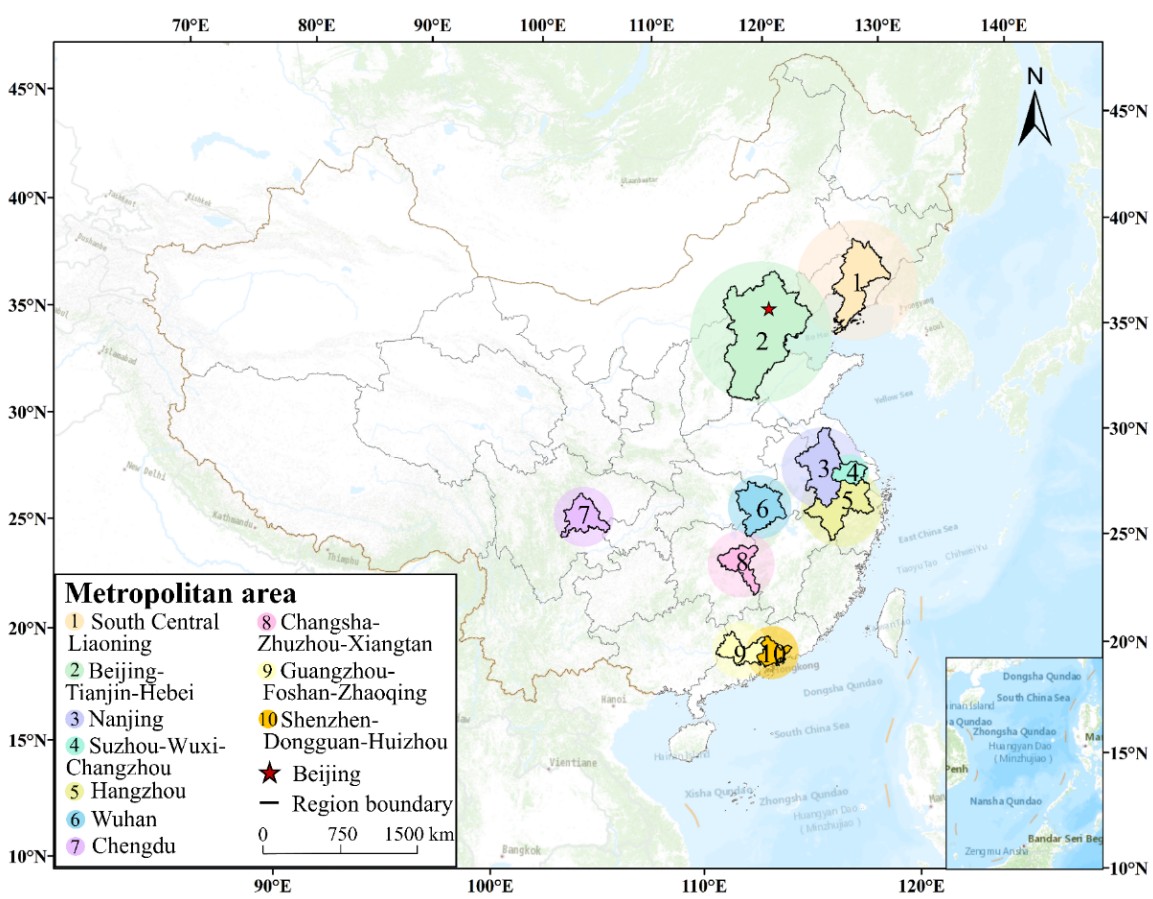

**Figure 1.** Distribution of ten metropolitan areas in China.

Cities in each metropolitan area are shown in Table 1. The South Central Liaoning metropolitan area is a region with an early industrial beginning and high urbanization level [41], and the Beijing-Tianjin-Hebei metropolitan area is the region with a high concentration of political, cultural and scientific, and technological activities in China [42,43]. Nanjing metropolitan area [44,45], Suzhou-Wuxi-Changzhou metropolitan area [46], and Hangzhou metropolitan area [47,48] are located in the urban agglomeration of the Yangtze River Delta. All of them play an important role in the network spatial pattern of "One core, Five circles, and Four belts" constructed by the urban agglomeration of the Yangtze River Delta. Wuhan metropolitan area [49–52] and Changsha-Zhuzhou-Xiangtan metropolitan

area [53] are important components of the Yangtze River Middle-Reach urban agglomeration. Chengdu metropolitan area is located in the core of the intersection, which is among the Belt and Road, Yangtze River Economic Belt, and the western land-sea new corridor, linking the east to the west and connecting the south to the north. Guangzhou-Foshan-Zhaoqing metropolitan area [54] and Shenzhen-Dongguan-Huizhou metropolitan area [55] are located within the urban agglomeration of the Pearl River Delta and play an important role in national economic construction.

**Table 1.** Cities in ten metropolitan areas.

| Metropolitan Areas | Cities |
|---|---|
| South Central Liaoning | Tieling, Shenyang, Fushun, Benxi, Liaoyang, Panjin, Anshan, Yingkou, Dalian |
| Beijing-Tianjin-Hebei | Chende, Zhangjiakou, Beijing, Tangshan, Qinhuangdao, Tianjin, Baoding, Langfang, Cangzhou, Shijiazhuang, Hengshui, Xingtai, Handan |
| Nanjing | Huaian, Yangzhou, Chuzhou, Zhenjiang, Ma'anshan, Wuhu, Xuancheng |
| Suzhou-Wuxi-Changzhou | Suzhou, Changzhou, Wuxi |
| Hangzhou | Jiaxing, Huzhou, Hangzhou, Shaoxing, Huangshan, Quzhou |
| Wuhan | Xiaogan, Huanggang, Wuhan, Ezhou, Huangshi, Xianning |
| Chengdu | Deyang, Chengdu, Ziyang, Meishan |
| Changsha-Zhuzhou-Xiangtan | Changsha, Zhuzhou, Xiangtan |
| Guangzhou-Foshan-Zhaoqing | Guangzhou, Foshan, Zhaoqing |
| Shenzhen-Dongguan-Huizhou | Shenzhen, Dongguan, Huizhou |

*2.2. Data Sources*

Research data are $PM_{2.5}$ data, meteorological data, and socioeconomic data from 2019 to 2020 which are shown in Table 2. The $PM_{2.5}$ data comes from the $PM_{2.5}$ historical data website [56]. The data provided by this website are the monthly average based on the hourly data of the China National Environmental Monitoring Station. In this paper, the $PM_{2.5}$ annual average is calculated based on the monthly average. By comparing station data with that in the database [57], it is found that some cities such as Nanjing (31 $\mu g/m^3$ for the former and 34.5 $\mu g/m^3$ for the latter) have a difference of about 3–5 $\mu g/m^3$, but some cities such as Beijing, Chengdu, and Changsha have a difference of about 6–9 $\mu g/m^3$. Since the database data have been processed twice, there are inevitably errors and missing values in this process, and the station data of direct use are more accurate, which is conducive to the study of this paper.

Socioeconomic data, including GDP, permanent population, population density, the proportion of secondary industry, the proportion of tertiary industry, and patent granted, are collected from the National Bureau of Statistics released by the China Statistical Yearbook [58] and statistical bulletins of each city [59]. Mobility [60] is also a pollution factor of $PM_{2.5}$, although this paper did not add mobility as a factor alone, GDP and population also have a connection to mobility. For example, Yoo [61] proposed that daily individual mobility and $PM_{2.5}$ exposure had a significant correlation. There are also economic and trade exchanges between cities. Thus, mobility is replaced by GDP and population in this paper. The meteorological data are the mean monthly maximum temperature, which comes from the China meteorological data network [62], and the mean annual maximum temperature is calculated according to the monthly average. Precipitation or wind-related indicators are important factors affecting $PM_{2.5}$ diffusion. However, some studies have shown that there is a strong U-shaped negative correlation between temperature and $PM_{2.5}$ concentration, which is lower in summer and autumn, and higher in spring and winter [63–65]. Precipita-

tion has a strong negative correlation with PM$_{2.5}$ in a small range, but the influence is not obvious in a long time and a large range [66]. For wind speed and wind direction in a short period of time, the reference force in small areas is stronger. Since this paper studies the annual scale, the correlation of the above indicators is relatively weak [67]. Therefore, this paper uses the average maximum temperature as the meteorological influencing factor.

**Table 2.** Data description.

| Data Type | Data Name | Unit | Description |
|---|---|---|---|
| Air quality | PM$_{2.5}$ | μg/m$^3$ | Describes the main object in the study |
| Economy | GDP | billion yuan | Describes the economic development of the city |
| Population | Permanent population<br>Population density | ten thousand people<br>person/km$^2$ | Describes the distribution of the urban population |
| Industry | The proportion of secondary industry<br>The proportion of tertiary industry | %<br>% | Describe the industrial structure of the city |
| Technology | Patent granted | piece | Describe the level of science and technology of the city |
| Meteorological | Mean annual maximum temperature | °C | - |

### 2.3. Methods

The modified gravity model was used to determine the PM$_{2.5}$ spatial correlation network matrix of ten metropolitan areas, and the matrix of each metropolitan area was standardized. The network density, network grade, and network efficiency were calculated to explore the overall characteristics of the PM$_{2.5}$ spatial correlation network. The centrality analysis was carried out by calculating the relative in-degree centrality, the relative out-degree centrality, the relative betweenness centrality, the relative in-closeness centrality, and the relative out-closeness centrality to explore the characteristics of each node in the network. Finally, QAP regression analysis was used to explore the influencing factors of the PM$_{2.5}$ spatial correlation network in ten metropolitan areas.

#### 2.3.1. Construction of the Spatial Correlation Network

The gravity model is usually used to construct the PM$_{2.5}$ spatial correlation network. The gravity model is a mathematical model based on Newton's law of universal gravitation, which is the basis of social network analysis and used to describe spatial interaction [27,68]. Therefore, the gravity model was selected to calculate PM$_{2.5}$ gravitational intensity in each city and construct the spatial correlation between nodes. In this paper, the geometric center of each city was used as a node, and the attribute data such as PM$_{2.5}$ and GDP were introduced into the improved gravity model to calculate gravity. The specific formula of the model is as follows:

$$F_{ij} = K_{ij} \frac{\sqrt[3]{P_i G_i M_i} \sqrt[3]{P_j G_j M_j}}{D_{ij}^2}, K_{ij} = \frac{V_i}{V_i + V_j} \tag{1}$$

where $i$, $j$ represent the city $i$ and city $j$. $P_i$, $G_i$, $M_i$ are the permanent population, GDP, and PM$_{2.5}$ concentration of city $i$, respectively. $D_{ij}$ is the shortest distance between city $i$ and city $j$. $K_{ij}$ is the weight. $V_i$ and $V_j$ are the PM$_{2.5}$ concentration of city $i$ and $j$, respectively. $F_{ij}$ means the gravitational intensity between city $i$ and city $j$.

Compared with the numerical value matrix, the relationship matrix can better reflect the relationship between cities in the PM$_{2.5}$ spatial correlation network [27]. Therefore, the average value of each row in the matrix was selected as the threshold. If the value in the matrix is higher than the threshold, it is defined as "1", indicating that there is a

relationship in PM$_{2.5}$ pollution between two cities. On the contrary, it is defined as "0", indicating that PM$_{2.5}$ pollution between two cities has no relationship.

### 2.3.2. Analysis of Overall Network Characteristics

Social network analysis (SNA) can better reflect the role of nodes in the network itself and the relationship between nodes by establishing an association network to analyze the overall network characteristics and the characteristics of each node [69,70]. Network density, network grade, and network efficiency reflect the overall characteristics of the PM$_{2.5}$ spatial correlation network. Network density reflects the tightness of the network. The greater the density, the stronger the spatial connection of cities [71,72], and its calculation formula is as follows:

$$ND = \frac{m}{n(n-1)} \tag{2}$$

where $ND$ is the network density. $m$ means the actual number of relationships in the network. $n$ is the number of nodes.

Network grade reflects the asymmetric accessibility of nodes [27]. The greater the network grade is, the more obvious the hierarchical structure among cities is. A few cities will be in a dominant position, and more cities will be subordinate to core cities, and its calculation formula is as follows:

$$NG = 1 - \frac{S}{\max(S)} \tag{3}$$

where $NG$ represents network grade. $S$ and max ($S$) are the actual and maximum number of pairs of cities symmetrically reachable in the network, respectively.

Network efficiency measures the degree of extra lines in the network. The lower the network efficiency is, the more stable the network structure is [73], and its calculation formula is as follows:

$$NE = 1 - \frac{R}{\max(R)} \tag{4}$$

where $NE$ is the network efficiency. $R$ is the number of extra lines. max ($R$) is the maximum possible number of extra lines.

### 2.3.3. Analysis of the Network Node Characteristics

SNA centrality analysis is a family of concepts for characterizing the structural importance of a node's position in a network. There are three indicators to measure the centrality in the network structure: the degree centrality, the betweenness centrality, and the closeness centrality [72]. Since the PM$_{2.5}$ spatial correlation network in this paper was a directed graph, the degree centrality was divided into in-degree centrality and out-degree centrality, and the closeness centrality was divided into in-closeness centrality and out-closeness centrality. However, when the scale of graphs is different, the local centrality of points in different graphs can not be compared horizontally, so the three indicators are all used by relativity [74]. The relative degree centrality is used to reflect the extent to which a city is influenced by other cities [75], and the relative betweenness centrality is an indicator to evaluate the location advantage of a city in the network. Relative closeness centrality is used to describe the extent to which a city is not controlled by other cities in the network [76], and the calculation formulas are as follows:

$$C_{RD}(i) = \frac{C_{AD}(i)}{n-1} \tag{5}$$

$$C_{RB}(i) = \frac{2}{(n-1)(n-2)} \sum_{j<k}^{k} r_{jk}(i)/r_{jk} \tag{6}$$

$$C_{RP}(i) = \frac{n-1}{\sum_{j=1}^{n} d_{ij}} \tag{7}$$

where $n$ is the number of nodes in the network. $C_{RD}(i)$ is the relative degree centrality of city $i$. $C_{AD}(i)$ is the number of other points connected with city $i$. $C_{RB}(i)$ is the relative betweenness centrality of city $i$. $r_{jk}$ is the number of shortcut distance that has a relationship between city $j$ and city $k$. $r_{jk}(i)$ is the number of shortcut distance passing through city $i$ between city $j$ and city $k$. $C_{RP}(i)$ is the relative closeness centrality of city $i$. $d_{ij}$ is the shortcut distance between city $i$ and city $j$.

### 2.3.4. Analysis of Influencing Factors

Because the PM$_{2.5}$ correlation matrix is a relationship matrix, there may be a high correlation between variables, which increases the standard deviation of parameter estimation. The quadratic assignment procedure (QAP) regression analysis is a method to obtain correlation coefficients between matrixes and conducts non-parametric tests on the coefficients through the random substitution in matrixes. The regression results obtained are adaptive, and the intercept and slope will change with the matrix value, so it can not be controlled artificially [77–79]. Therefore, this paper uses QAP regression analysis to explore the relationship between the PM$_{2.5}$ correlation matrix and influencing factor matrixes:

$$M = \mathrm{f}(L, S, T, P, A, H) \tag{8}$$

where $M$ represents the PM$_{2.5}$ spatial correlation matrixes in metropolitan areas. $L$ means the geographical locational relationship matrixes between cities within the metropolitan area, the adjacent cities are marked as '1', and the non-adjacent cities are marked as '0'. $S$ is the secondary industrial proportion differences' matrixes. $T$ is the tertiary industrial proportion differences matrixes. $P$ is the population density differences matrixes. $A$ is the patent granted differences matrixes. $H$ is the mean annual maximum temperature differences matrixes.

## 3. Results

### 3.1. Overall Network Characteristics

The gravity model was used to calculate the spatial correlation matrix of PM$_{2.5}$ in ten metropolitan areas, and the number of nodes and relationships were shown in Table 3. Compared with the results of two years, the number of relationships among Nanjing metropolitan area, South Central Liaoning metropolitan area, and Chengdu metropolitan area decreased slightly. Beijing-Tianjin-Hebei metropolitan area, Hangzhou metropolitan area, Wuhan metropolitan area, Guangzhou-Foshan-Zhaoqing metropolitan area, Shenzhen-Dongguan-Huizhou metropolitan area, Suzhou-Wuxi-Changzhou metropolitan area, and Changsha-Zhuzhou-Xiangtan metropolitan area did not change significantly.

**Table 3.** The number of nodes and relationships in ten metropolitan areas from 2019 to 2020.

| Metropolitan Areas | The Number of Nodes | The Number of Relationships | |
|---|---|---|---|
| | | 2019 | 2020 |
| Beijing-Tianjin-Hebei | 13 | 45 | 45 |
| Nanjing | 8 | 17 | 16 |
| Hangzhou | 6 | 12 | 12 |
| Wuhan | 6 | 8 | 8 |
| South Central Liaoning | 9 | 26 | 23 |
| Chengdu | 4 | 5 | 4 |
| Guangzhou-Foshan-Zhaoqing | 3 | 3 | 3 |
| Shenzhen-Dongguan-Huizhou | 3 | 3 | 3 |
| Suzhou-Wuxi-Changzhou | 3 | 3 | 3 |
| Changsha-Zhuzhou-Xiangtan | 3 | 3 | 3 |

The network relationship of PM$_{2.5}$ in each metropolitan area was better displayed in Figure 3. The network structure of the Beijing-Tianjin-Hebei metropolitan area, Nanjing

metropolitan area, Hangzhou metropolitan area, Wuhan metropolitan area, and South Central Liaoning metropolitan area was more complex than other metropolitan areas due to the number of nodes and relationships.

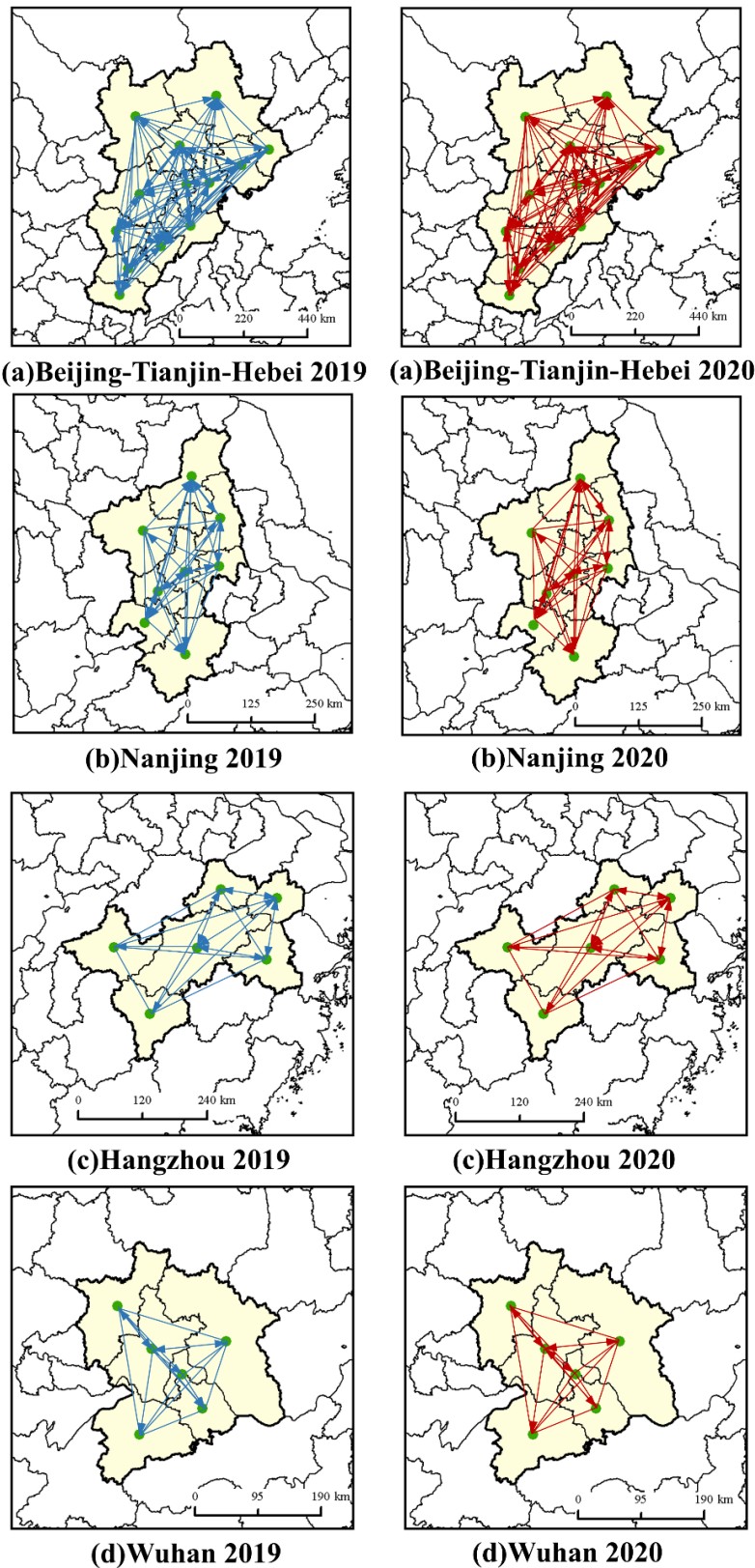

**Figure 2.** *Cont*.

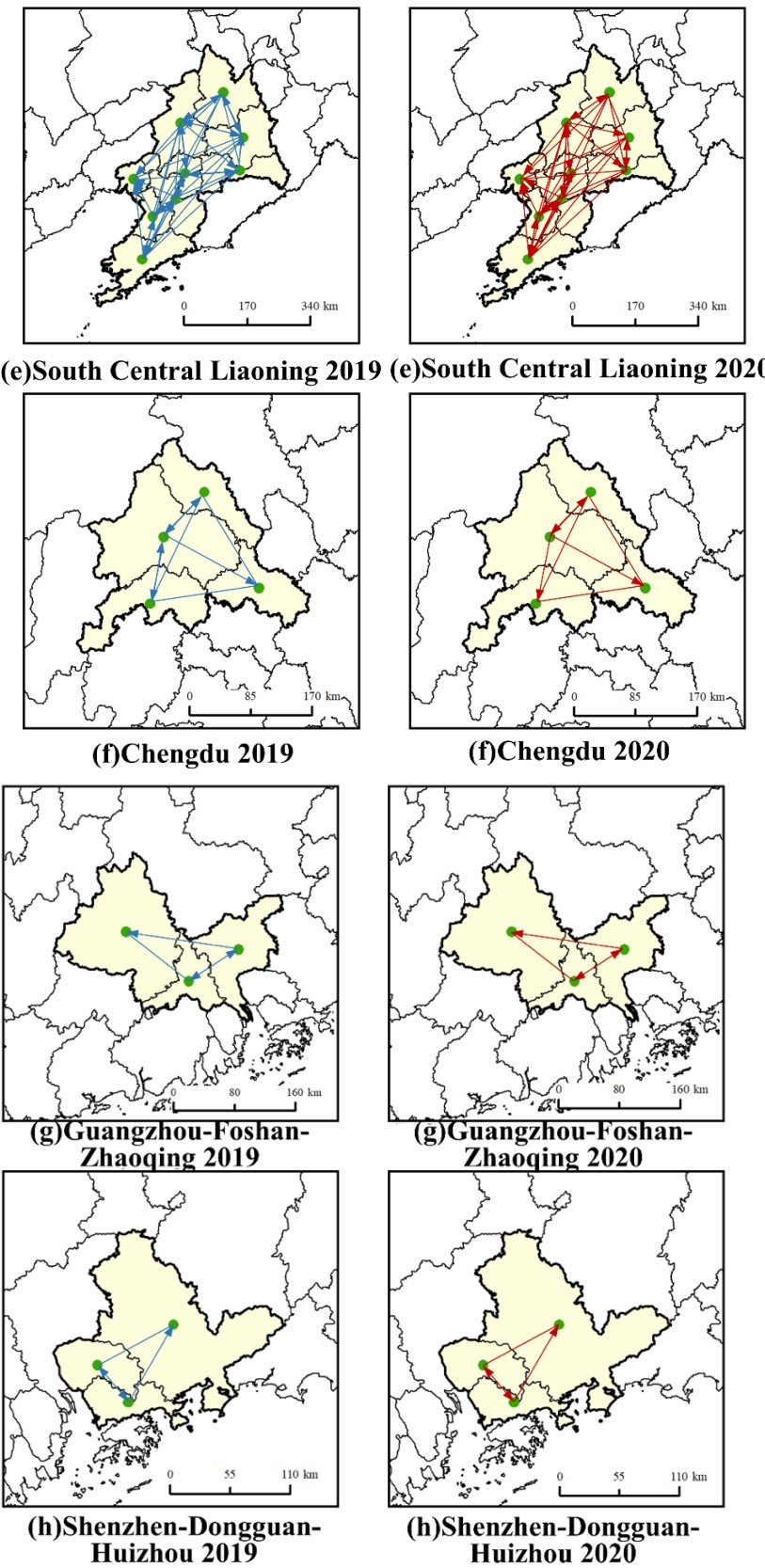

**(e)South Central Liaoning 2019 (e)South Central Liaoning 2020**

**(f)Chengdu 2019**                    **(f)Chengdu 2020**

**(g)Guangzhou-Foshan-Zhaoqing 2019**     **(g)Guangzhou-Foshan-Zhaoqing 2020**

**(h)Shenzhen-Dongguan-Huizhou 2019**     **(h)Shenzhen-Dongguan-Huizhou 2020**

**Figure 3.** *Cont.*

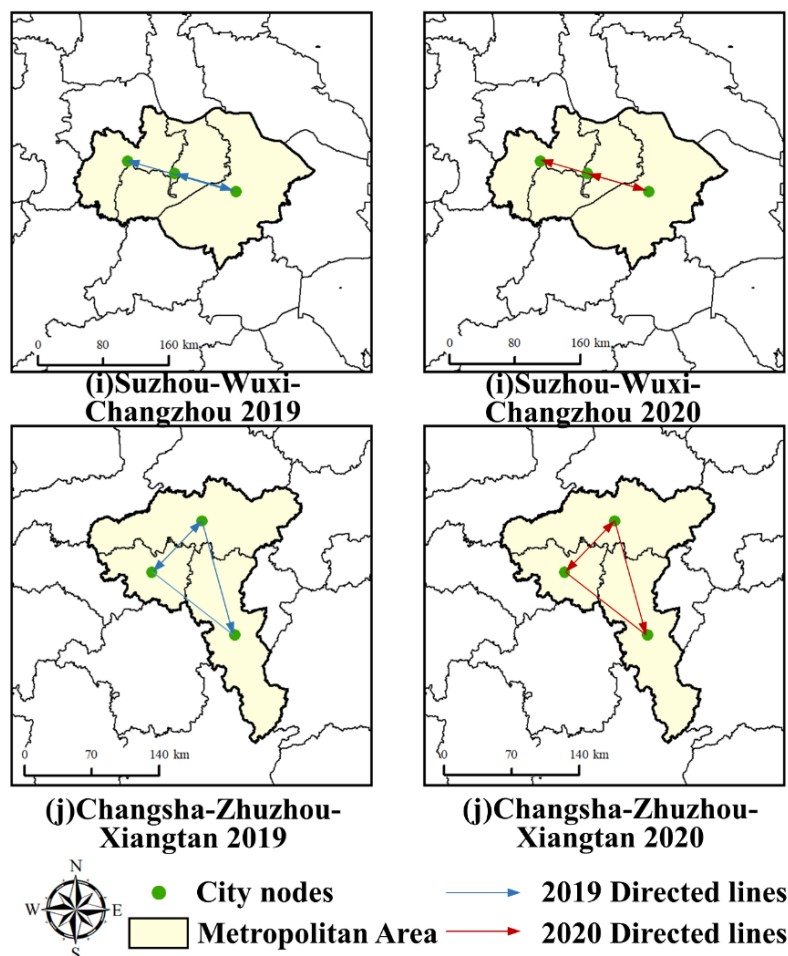

**Figure 3.** PM$_{2.5}$ spatial correlation networks in ten metropolitan areas from 2019 to 2020 (arrows indicate relationships between the two cities).

Considering the different number of nodes in metropolitan areas, the spatial comparison of the network characteristics of each metropolitan area may have an impact. Because there was no metropolitan area composed of five cities, this paper was based on the criteria which took more than five cities as large metropolitan areas and less than five cities as small metropolitan areas; the ten metropolitan areas were divided into two categories: the Beijing-Tianjin-Hebei metropolitan area, Nanjing metropolitan area, Hangzhou metropolitan area, Wuhan metropolitan area, and South Central Liaoning metropolitan area were the first category. Chengdu metropolitan area, Guangzhou-Foshan-Zhaoqing metropolitan area, Shenzhen-Dongguan-Huizhou metropolitan area, Suzhou-Wuxi-Changzhou metropolitan area, and Changsha-Zhuzhou-Xiangtan metropolitan area were the second category.

By comparing the characteristics of the PM$_{2.5}$ spatial correlation network in 2019 and 2020, the three main points are as follows:

(1) From the perspective of network density, the Nanjing metropolitan area, South Central Liaoning metropolitan area, Chengdu metropolitan area, and Shenzhen-Dongguan-Huizhou metropolitan area were decreased, while the other three metropolitan areas did not change. In the first category, the network density of the Hangzhou metropolitan area and South Central Liaoning metropolitan area were higher than that of other metropolitan areas, which ranged from 0.31 to 0.4. This result showed that the characteristic industry is an important factor to promote connection in metropolitan areas. The tertiary industry of the Hangzhou metropolitan area such as the Yiwu small commodity market in Zhejiang and early developed industries and industrial activities within cities in the South Central Liaoning metropolitan area. However, the network density of the Wuhan metropolitan

area was 0.267, which was the lowest. In the second category, the network density of the Chengdu metropolitan area was the lowest (0.417 and 0.333, respectively), and the other four metropolitan areas were high (0.5–0.75). This paper argues that there are several reasons for the high network density of metropolitan areas. On the one hand, the Suzhou-Wuxi-Changzhou metropolitan area has similar industrial structures among cities. On the other hand, advantageous geographical location and convenient transportation also play a role. In particular, the Guangzhou-Foshan-Zhaoqing metropolitan area and the Shenzhen-Dongguan-Huizhou metropolitan area take the development line from Guangzhou to Zhuhai and Guangzhou to Shenzhen as the spindle through the Hong Kong-Zhuhai-Macao Bridge. In addition, the decrease in network density indicated that the number of relationships between cities was decreased, and the economic, population and $PM_{2.5}$ transmission connection among cities were all affected by COVID-19. Therefore, every region should attach importance to environmental protection and investment, strengthen cooperation, and improve network density after the epidemic was relieved (Figure 4).

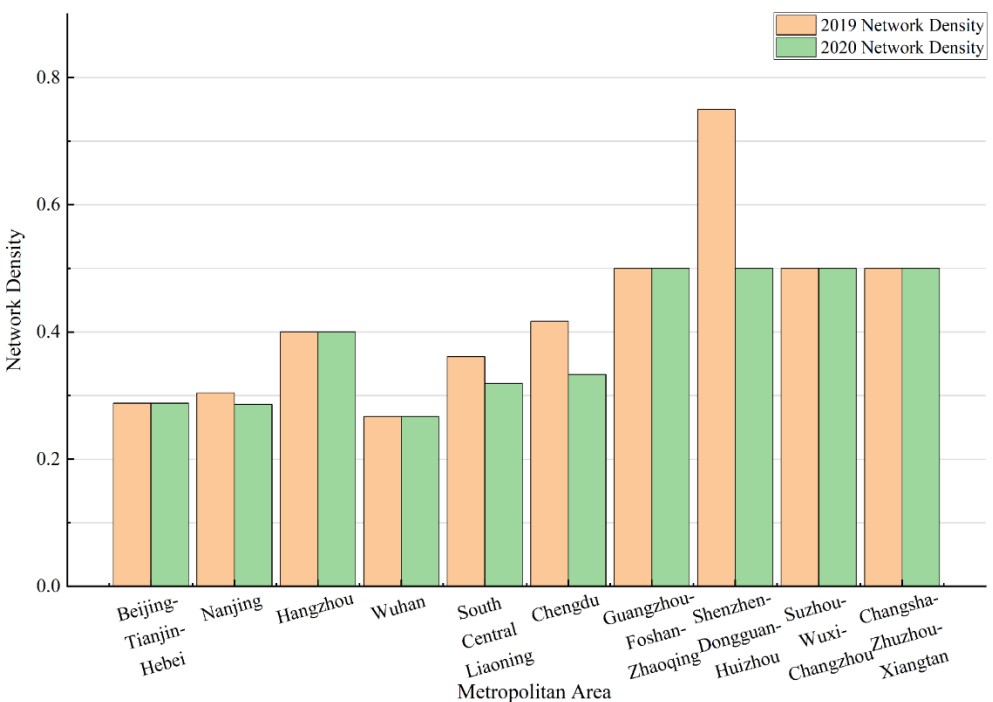

**Figure 4.** Network density of $PM_{2.5}$ spatial correlation networks in ten metropolitan areas in 2019–2020.

(2) From the perspective of network grade, the first category of the metropolitan area was low except for the Wuhan metropolitan area and Hangzhou metropolitan area, and the second category of the metropolitan area was all high. In the first category, except for the decrease in the Wuhan metropolitan area and the unchanged Hangzhou metropolitan area, the rest of the metropolitan areas had increased. The network grade of the Wuhan metropolitan area and Hangzhou metropolitan area were higher than 0.5, indicating that their hierarchical structure was strict. However, the Beijing-Tianjin-Hebei metropolitan area, South Central Liaoning metropolitan area, and Nanjing metropolitan area had the weak hierarchical structure, and $PM_{2.5}$ pollution in each city had a strong influence on each other. In the second category, the network grade of the Chengdu metropolitan area increased from 0.5 to 0.8 in two years. While the other four metropolitan areas were high and the change was stable, both of which were 0.667, it indicated that they had a strict hierarchical structure, and some cities in the dominant position and more cities in the edge position. To explore the reasons, this paper thought that the Hangzhou metropolitan area, Guangzhou-Foshan-Zhaoqing metropolitan area, Shenzhen-Dongguan-Huizhou metropolitan area, Suzhou-

Wuxi-Changzhou metropolitan area, and Changsha-Zhuzhou-Xiangtan metropolitan area took the core cities as the main spindle to promote the development of the surrounding cities. Therefore, the PM$_{2.5}$ spatial correlation networks in these metropolitan areas had a strict and stably changing hierarchical structure (Figure 5).

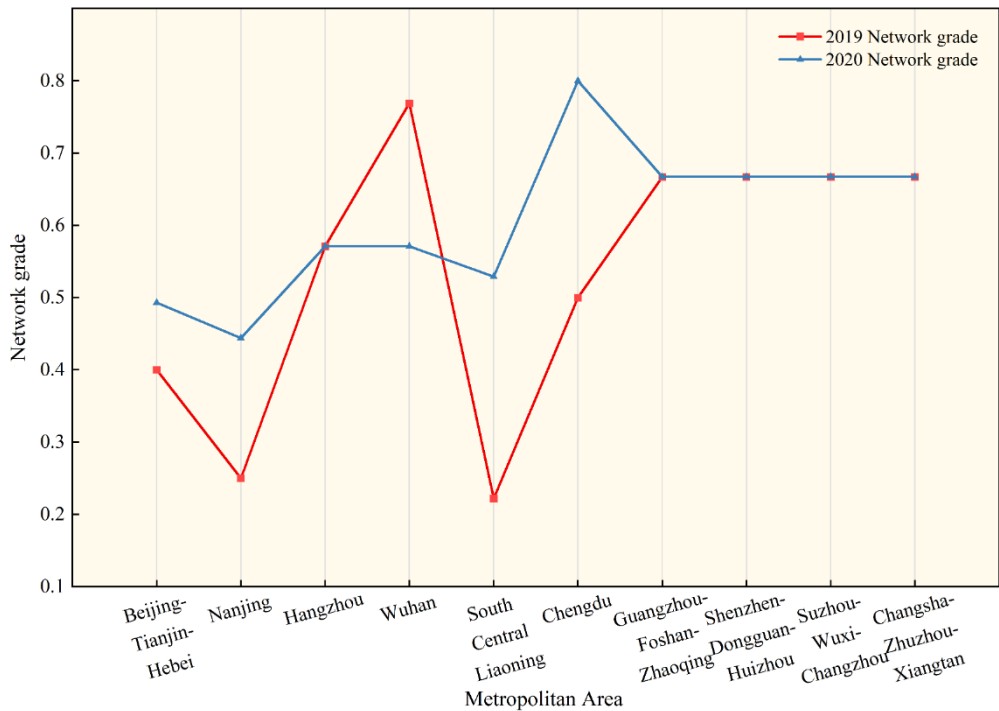

**Figure 5.** Network grade of PM$_{2.5}$ spatial correlation networks in ten metropolitan areas in 2019–2020.

(3) From the perspective of network efficiency, both the first category and the second category metropolitan areas were between 0.6 and 1, indicating that the PM$_{2.5}$ spatial correlation network structure of the ten metropolitan areas was not stable, and the networks would be damaged due to the interruption of the relationship between nodes (Figure 6).

This paragraph focused on the analysis of the Wuhan metropolitan area due to the outbreak of COVID-19 in 2019–2020; Wuhan was the birthplace of COVID-19 in China, which had a great impact on the economic development of Wuhan. Both the population and GDP declined, and thus the stability of the network hierarchical structure in the Wuhan metropolitan area decreased and network efficiency fluctuated greatly. Although COVID-19 also had an impact on other metropolitan areas, the GDP and population of some cities still increased slightly, so the network grade increased and the network efficiency fluctuated little [7].

### 3.2. Centrality Analysis

Based on the five centrality indicators of each node, the relative out-degree centrality ($X_1$), the relative in-degree centrality ($X_2$), the relative betweenness centrality ($X_3$), the relative out-closeness centrality ($X_4$), and the relative in-closeness centrality ($X_5$), then the mean value of each indicator in the metropolitan area from 2019 to 2020 was calculated as a standard to explore the status and role of each node in the network (Table 4).

This paper selected Beijing-Tianjin-Hebei metropolitan area to amplify node characteristics. It could be seen from Figure 7 that the relative out-degree centrality ($X_1$) and relative in-degree centrality ($X_2$) of Beijing ($m_3$), Tianjin ($m_6$), Baoding ($m_7$), and Shijiazhuang ($m_{10}$) were all higher than the mean values of the metropolitan area. This indicated that these cities had a strong ability to absorb and output PM$_{2.5}$ in the network. Two cities had a stronger ability to absorb PM$_{2.5}$ than to export in the network such as Tangshan ($m_4$) and Langfang ($m_8$). There were also cities with greater PM$_{2.5}$ output ability than absorption

ability, such as Cangzhou ($m_9$) and Hengshui ($m_{11}$). The rest of the cities such as Chengde ($m_1$) and Zhangjiakou ($m_2$) in the Beijing-Tianjin-Hebei metropolitan area had the ability to absorb and output $PM_{2.5}$ was all weak in the network.

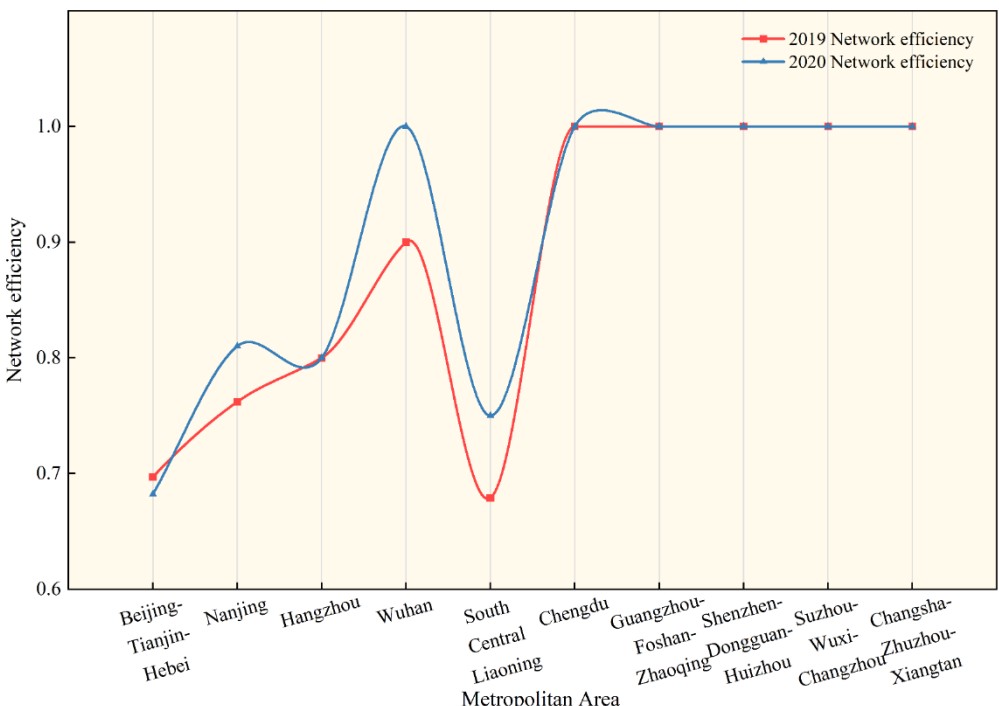

**Figure 6.** Network efficiency of $PM_{2.5}$ spatial correlation networks in ten metropolitan areas in 2019–2020.

**Table 4.** The mean values of five centrality indicators in ten metropolitan areas from 2019 to 2020.

| Metropolitan Areas | $X_1$ | $X_2$ | $X_3$ | $X_4$ | $X_5$ |
|---|---|---|---|---|---|
| Beijing-Tianjin-Hebei | 28.85 | 28.85 | 6.70 | 20.91 | 42.66 |
| Nanjing | 29.46 | 29.46 | 15.48 | 31.05 | 46.09 |
| Hangzhou | 40.00 | 40.00 | 6.67 | 34.35 | 54.96 |
| Wuhan | 26.67 | 26.67 | 8.33 | 30.97 | 47.04 |
| South Central Liaoning | 34.03 | 34.03 | 6.37 | 33.12 | 48.08 |
| Chengdu | 37.50 | 37.50 | 12.50 | 43.51 | 56.88 |
| Guangzhou-Foshan-Zhaoqing | 50.00 | 50.00 | 16.67 | 55.56 | 66.67 |
| Shenzhen-Dongguan-Huizhou | 50.00 | 50.00 | 16.67 | 55.56 | 66.67 |
| Suzhou-Wuxi-Changzhou | 50.00 | 50.00 | 16.67 | 55.56 | 66.67 |
| Changsha-Zhuzhou-Xiangtan | 50.00 | 50.00 | 16.67 | 55.56 | 66.67 |

It could be seen from the results of relative betweenness centrality ($X_3$) that Beijing ($m_3$), Tianjin ($m_6$), Baoding ($m_7$), and Shijiazhuang ($m_{10}$) were all higher than the mean value of the metropolitan area, indicating that these cities had a strong ability to control the transfer of $PM_{2.5}$ among other cities in the network. They played a "bridge" role in the process of $PM_{2.5}$ transfer and were the most important media cities in the spatial correlation network of $PM_{2.5}$ in each metropolitan area. The relative betweenness centrality ($X_3$) of the other cities was low, at the edge of the network.

Except for Xingtai ($m_{12}$), Handan ($m_{13}$), and Qinhuangdao ($m_5$), cities in the Beijing-Tianjin-Hebei metropolitan area had a relative out-closeness centrality ($X_4$) and relative in-closeness centrality ($X_5$) higher than the mean value of the metropolitan area, especially Beijing ($m_3$) and Tianjin ($m_6$), which were up to 70 or more, indicating that these cities had a strong ability to control the output and emission of $PM_{2.5}$ in the network, and they were important nodes to maintain contact with other cities in the network. They had important

demonstration value in establishing the regional joint pollution prevention mechanism of $PM_{2.5}$.

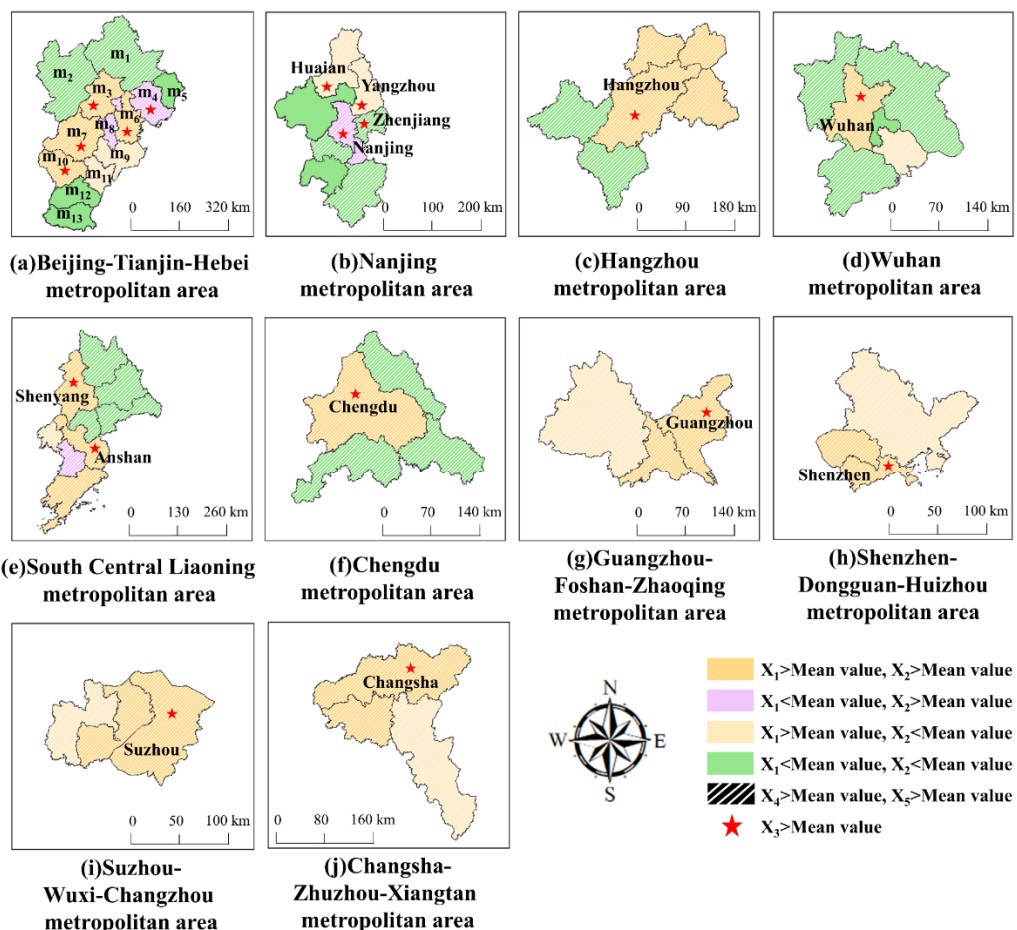

**Figure 7.** The centrality analysis results of $PM_{2.5}$ spatial correlation in ten metropolitan areas from 2019 to 2020. Note: $m_1$ is Chengde, $m_2$ is Zhangjiakou, $m_3$ is Beijing, $m_4$ is Tangshan, $m_5$ is Qinhuangdao, $m_6$ is Tianjin, $m_7$ is Baoding, $m_8$ is Langfang, $m_9$ is Cangzhou, $m_{10}$ is Shijiazhuang, $m_{11}$ is Hengshui, $m_{12}$ is Xingtai, $m_{13}$ is Handan.

According to the similarities and differences of the above five indicators in metropolitan areas, this paper found that the northern metropolitan areas were "multi-core" metropolitan areas, such as the Beijing-Tianjin-Hebei metropolitan area and South Central Liaoning metropolitan area. There were usually several core cities in these kinds of metropolitan areas. For instance, Beijing ($m_3$), Tianjin ($m_6$), Baoding ($m_7$), and Shijiazhuang ($m_{10}$) were the core cities of the Beijing-Tianjin-Hebei metropolitan area, whose five indicators were all ahead of other cities. Similarly, Shenyang and Anshan led the South Central Liaoning metropolitan area. In addition, the southern metropolitan areas were "single-core" metropolitan areas, such as Hangzhou metropolitan area, Suzhou-Wuxi-Changzhou metropolitan area, and eight other metropolitan areas. Such metropolitan areas usually had the only core city, and the five indicators were far ahead of other cities. Moreover, cities in the core position of metropolitan area networks were prominent in industry, economy, and population. For example, Shenyang, Anshan, etc. in the South Central Liaoning metropolitan area were cities with developed heavy industry, and they could be more closely linked with the other cities. Hangzhou metropolitan area, Nanjing metropolitan area, etc. were among the economies of cities that were developed. They could also have a greater connection with other cities. Cities have different roles and responsibilities in the $PM_{2.5}$ spatial correlation network. Therefore, it is particularly important for govern-

ments to control PM$_{2.5}$ pollution according to local conditions rather than following the general trend.

### 3.3. Analysis of Influencing Factors of Spatial Correlation

In the previous studies, geographical locational relationship (*L*), the secondary industrial proportion differences (*S*), the tertiary industrial proportion differences (*T*), the mean annual maximum temperature differences (*H*), and population density differences (*P*) were often selected as the influencing factors of the pollution correlation network [27,55]. However, core cities in the metropolitan area tended to have strong technological innovation capacity, which also attracted edge cities to cooperate with them, and the correlation between cities was enhanced. Therefore, in this paper, the patent granted differences (*A*) was used as a factor to expand the influencing factor system for QAP regression analysis to study the influencing factors of spatial correlation of PM$_{2.5}$ in metropolitan areas. The results were shown in Table 5.

Considering the results of 2019 and 2020, the geographical locational relationship (*L*) had a significant impact on the Beijing-Tianjin-Hebei metropolitan area, Nanjing metropolitan area, Hangzhou metropolitan area, and the South Central Liaoning metropolitan area. The regression coefficient was positive and had passed 1% or 5% of the significant test; in particular, the regression coefficient of the Nanjing metropolitan area was more than 0.5. It indicated that the proximity relationship between cities in the metropolitan area can promote connections between regions. The secondary industrial proportion differences (*S*) and the tertiary industrial proportion differences (*T*) had both positive and negative impacts on the PM$_{2.5}$ correlation intensity in the metropolitan area. For example, the secondary industrial proportion differences (*S*) were positively correlated with the PM$_{2.5}$ correlation intensity in the Beijing-Tianjin-Hebei metropolitan area, and the expansion of the secondary industry gap could facilitate the exchange of industrial activities between cities and thus promote the connection in the metropolitan area. The mean annual maximum temperature differences (*H*) had a negative impact on the metropolitan areas such as the Beijing-Tianjin-Hebei metropolitan area and Chengdu metropolitan area. They passed the 5% or 10% significance test. The impact of population density differences (*P*) on the PM$_{2.5}$ network was not only positive but also negative. For example, the correlation of the impact on the Chengdu metropolitan area had changed significantly in two years, indicating that its impact on the connection between cities in Chengdu metropolitan area was unstable, and the population flow between cities should be reasonably controlled. The patent granted differences (*A*) had a significant impact on most metropolitan areas and the correlation was positive, such as the Beijing-Tianjin-Hebei metropolitan area and Nanjing metropolitan area, which were significant at the 1% level. It meant that the greater the difference in science and technology between cities, the more cooperation and exchange could be promoted.

To sum up, the top three influencing factors of the Beijing-Tianjin-Hebei metropolitan area, Nanjing metropolitan area, Hangzhou metropolitan area, Wuhan metropolitan area, South Central Liaoning metropolitan area, and Chengdu metropolitan area were geographical locational relationship (*L*), the secondary industrial proportion differences (*S*), and patent granted differences (*A*), but the Guangzhou-Foshan-Zhaoqing metropolitan area, Shenzhen-Dongguan-Huizhou metropolitan area, Suzhou-Wuxi-Changzhou metropolitan area, and Changsha-Zhuzhou-Xiangtan metropolitan area had no dominant influencing factors.

Based on the above results, this paper argues that, although the geographical locational relationship is inevitable, local governments can regularly carry out exchanges, reach a certain degree of cooperation and consensus, and overcome the problems caused by geographical locational differences to the greatest extent. Secondly, spatial correlation between cities can be strengthened by adjusting the industrial structure, controlling population mobility, and strengthening scientific and technological exchanges, which also prove the effectiveness and importance of controlling PM$_{2.5}$ according to local conditions in various regions.

**Table 5.** QAP regression results for 2019 and 2020.

| Metropolitan Areas | 2019 | | | | | | 2020 | | | | | |
|---|---|---|---|---|---|---|---|---|---|---|---|---|
| | **L** | **S** | **T** | **H** | **P** | **A** | **L** | **S** | **T** | **H** | **P** | **A** |
| Beijing-Tianjin-Hebei | 0.398 *** | 0.155 * | −0.295 ** | −0.182 ** | 0.063 | 0.435 *** | 0.390 *** | 0.235 ** | −0.351 ** | −0.185 ** | 0.013 | 0.491 *** |
| Nanjing | 0.543 *** | 0.197 | −0.785 ** | −0.005 | −0.205 | 0.044 *** | 0.564 *** | 0.099 | −0.007 *** | 0.010 | 0.064 | 0.094 *** |
| Hangzhou | 0.332 ** | −0.395 * | 0.928 ** | −0.438 * | 0.316 * | 0.385 *** | 0.718 *** | −0.009 | 0.322 | −0.047 | 0.155 | 0.251 * |
| Wuhan | 0.243 ** | −0.371 *** | 0.074 | 0.001 | −0.328 * | 0.860 | 0.073 | −0.177 ** | −0.296 | 0.001 | −0.131 | 0.223 ** |
| South Central Liaoning | 0.736 *** | −0.081 | −0.045 | 0.039 | −0.010 | 0.165 * | 0.484 *** | −0.212 ** | 0.245 * | 0.020 | −0.350 ** | 0.379 ** |
| Chengdu | 0.104 | −0.064 | 0.818 | 0.049 | 0.225 ** | −0.347 ** | −0.240 | 0.572 * | −0.654 | −0.151 ** | −0.151 ** | 0.882 ** |
| Guangzhou-Foshan-Zhaoqing | 0.001 | 0.231 | −0.001 | 0.001 | −0.233 | 0.485 | 0.001 | 0.208 | −0.003 | 0.001 | −0.233 | 0.546 |
| Shenzhen-Dongguan-Huizhou | 0.001 | 0.567 | 0.000 | 0.000 | 0.112 | −0.011 | 0.001 | 0.384 | −0.001 | 0.001 | −0.263 | 0.493 |
| Suzhou-Wuxi-Changzhou | 0.056 | −0.001 | −0.048 | −0.029 | −0.138 | 0.619 | 0.001 | −0.001 | −0.000 | 0.001 | −0.216 | 0.621 |
| Changsha-Zhuzhou-Xiangtan | 0.001 | 0.081 | 0.083 | 0.001 | −0.208 | 0.674 | 0.001 | 0.160 | 0.125 | 0.009 | −0.330 | 0.714 |

Note: *** means significant at 1% level, ** means significant at 5% level, * means significant at 10% level.

According to the similarities and differences of influencing factors and the annual average economic development of metropolitan areas in two years (Table 6), this paper found that three metropolitan areas were at the "mature stage of economic development", such as the Suzhou-Wuxi-Changzhou metropolitan area, Guangzhou-Foshan-Zhaoqing metropolitan area, and Shenzhen-Dongguan-Huizhou metropolitan area, and their GDP all reached 10 trillion yuan, and the patent granted exceeded most other cities. Moreover, all factors had no significant impact on them, indicating that the economic development level of cities in these metropolitan areas was relatively balanced. Secondly, four metropolitan areas were at the "growth stage of economic development", such as the South Central Liaoning metropolitan area, Beijing-Tianjin-Hebei metropolitan area, Hangzhou metropolitan area, and the Nanjing metropolitan area. They were mainly affected by geographical location, industrial structure, and technological level. These metropolitan areas were currently in the growth period of economic development. In the later period, the geographical location advantages should be used to promote overseas trade of coastal cities and strengthen urban cooperation. Finally, the remaining three metropolitan areas were at the "reform stage of economic development". They were less affected by factors and their economic development characteristics were not prominent. In the future, governments should focus on the development of characteristic industries and promote urban exchanges and cooperation.

**Table 6.** The average annual economic growth of ten metropolitan areas in 2019–2020.

| Metropolitan Areas | GDP (Billion Yuan) | The Proportion of Secondary Industry (%) | The Proportion of Tertiary Industry (%) | Patent Granted (Piece) |
|---|---|---|---|---|
| Beijing-Tianjin-Hebei | 6532.63 | 34.20 | 55.72 | 22,323 |
| Nanjing | 4860.61 | 45.5 | 48.51 | 17,981 |
| Hangzhou | 5453.82 | 42.62 | 53.13 | 12,233 |
| Wuhan | 4125 | 41.93 | 47.22 | 10,591 |
| South Central Liaoning | 2335.25 | 41.72 | 49.21 | 4757 |
| Chengdu | 5482.32 | 37.11 | 50.96 | 16,194 |
| Guangzhou-Foshan-Zhaoqing | 12,462.6 | 41.08 | 52.06 | 46,982 |
| Shenzhen-Dongguan-Huizhou | 13,688.2 | 48.25 | 49.93 | 92,893 |
| Suzhou-Wuxi-Changzhou | 13,139.2 | 46.99 | 51.64 | 49,140 |
| Changsha-Zhuzhou-Xiangtan | 5737.73 | 45.58 | 48.78 | 12,543 |

## 4. Discussion

This study aims to explore the spatial effect of $PM_{2.5}$ pollution from a network perspective, analyze the current situation of $PM_{2.5}$ coordinated control in China's ten metropolitan areas, and determine the factors affecting the $PM_{2.5}$ spatial association network.

Firstly, the annual average $PM_{2.5}$ reflects the overall level, which is used for macroscale research, which caters to the time scale of this study. In local regions, $PM_{2.5}$ changes dramatically due to weather, seasonality of economic activities. Thus, it is suitable to use fine-grained $PM_{2.5}$ data to study.

Secondly, in the overall network characteristics, we found that the network density of four metropolitan areas decreased, compared with Su [27] of each policy carried out after the initial network density peak began to decline. However, in this paper, the time range is from 2019 to 2020, but since the "13th Five-Year Plan", China has paid more attention to environmental protection. Therefore, excluding policy factors, the impact of the epidemic on links between cities can not be ignored. The results of network grade in this paper also verified the phenomenon that the network grade of various regions in China showed an overall upward trend by 2015 mentioned by Su's study.

Thirdly, in the centrality analysis, from the five centrality indicators, it could be concluded that the status and role of cities in the network in each metropolitan area were different and could be divided into "single-core" and "multi-core" metropolitan areas. This result can be verified by Song's [68] centrality analysis of Chengdu-Chongqing urban agglomeration. However, due to the large coverage of this paper, by comparing the

similarity and difference of centrality results in different metropolitan areas, the conclusion of "single-core" and "multi-core" is more universal.

Finally, geographical location, secondary industrial activities, and the level of science and technology were the main factors. This result can be verified by Wu's [79] discovery. However, a scientific factor was added to enrich the factor system, which is also different from his discovery. In addition, this study found three kinds of phenomena in economic development stages from the results, which is also different from other papers.

From what has been discussed above, $PM_{2.5}$ pollution control is not only related to a single city but also the establishment of a long-term and coordinated $PM_{2.5}$ control mechanism for the whole region. Therefore, this paper puts forward some suggestions for the control of $PM_{2.5}$ pollution in China's ten metropolitan areas. On the one hand, government departments should carefully track $PM_{2.5}$ pollution sources and transmission channels by controlling the core cities in the $PM_{2.5}$ spatial correlation network, such as Chengdu, Hangzhou, Shenyang, Nanjing, and Wuhan, and exerting their influence on $PM_{2.5}$ pollution to drive and guide other cities to control effectively. In particular, government departments should consider the location of cities in the network and their spatial spillover effects when formulating policies to curb $PM_{2.5}$ pollution, so as to adjust measures to local conditions. On the other hand, cities should assume different responsibilities in $PM_{2.5}$ pollution control. For example, $PM_{2.5}$ pollution control funds should be established to compensate cities affected by pollution which comes from cities with a large proportion of secondary industry in their industrial structure. In addition, significant influence on $PM_{2.5}$ pollution factor can be used by each city. For example, $PM_{2.5}$ pollution can be reduced by optimizing the industrial structure, population structure, energy structure, the development of new technology, and new energy. Using the relationship of the geographical location and technology differences between cities, communication and contact with cities can be established actively to narrow differences in overall development levels and deepen regional cooperation.

In this study, the contribution ability of each city in the $PM_{2.5}$ spatial correlation network was explored in-depth, and corresponding control suggestions of $PM_{2.5}$ pollution were put forward according to the output or input capacity of $PM_{2.5}$. The research results can improve the adaptation of $PM_{2.5}$ pollution control policies to local conditions and moderate the differences in the comprehensive development level of each city. Regional cooperation will be deepened to promote coordinated control of $PM_{2.5}$ pollution. However, due to the limitation of time granularity of statistical yearbook data, the number of influencing factors in QAP regression analysis is insufficient, which is the direction to be improved on in subsequent experiments. Future research will focus on eliminating the loss of sequence rules caused by the data averaging process, improving the accuracy of $PM_{2.5}$ spatial modeling, and supplementing the $PM_{2.5}$ concentration prediction model to control air pollution.

## 5. Conclusions

The social network analysis method was adopted to establish the $PM_{2.5}$ spatial correlation network of ten Chinese metropolitan areas based on the gravity model in this study. The characteristics of the overall network and each node were analyzed. Finally, the QAP regression analysis method was used to explore the influencing ability of each factor on the $PM_{2.5}$ spatial correlation network. The main conclusions were as follows: (1) The network density of half of the metropolitan areas showed a decreasing trend during 2019–2020, while the other metropolitan areas showed no change. The network density of the Wuhan and Chengdu metropolitan area was the lowest, but the network density of the Hangzhou metropolitan area and the other five metropolitan areas was higher than the rest. Secondly, six metropolitan areas such as Wuhan metropolitan area had high network grade, but the network structure of the ten metropolitan areas was not stable. (2) The spatial correlation network of $PM_{2.5}$ in ten metropolitan areas showed trends of centralization and marginalization. The Beijing-Tianjin-Hebei metropolitan area and the South Central Liaoning metropolitan area were "multi-core" metropolitan areas, and the other eight

metropolitan areas were "single-core" pattern. (3) The top three influencing factors for six metropolitan areas were the geographical locational relationship, the secondary industrial proportion differences, and patent granted differences. However, six factors had no significant influence on the $PM_{2.5}$ spatial correlation network in the other four metropolitan areas. According to the commonness and differences in influencing factors results and economic conditions, the ten metropolitan areas could be divided into three categories: "mature stage of economic development", "growth stage of economic development", and "reform stage of economic development".

This study reveals the importance of coordinated $PM_{2.5}$ pollution control, which can help local governments make policies to control $PM_{2.5}$ pollution according to local conditions. Clarifying the status and role of each city in the network can achieve coordinated economic and ecological development and conform to the trend of sustainable development.

**Author Contributions:** Conceptualization, Shuaiqian Zhang. and Tong Zhou; methodology, Shuaiqian Zhang and Qi Wu; software, Shuaiqian Zhang; resources, Qi Wu, Yu Wang and Qile Han; data curation, Shuaiqian Zhang and Qi Wu; writing—original draft preparation, Shuaiqian Zhang; writing—review and editing, Fei Tao and Tong Zhou; visualization, Qile Han, Yu Wang, and Qi Wu; funding acquisition, Fei Tao and Tong Zhou. All authors have read and agreed to the published version of the manuscript.

**Funding:** This research was funded by the Major project of the National Social Science Fund under Grant No. 19ZDA189; in part by the Natural Science and Technology Project of Nantong (MS12020075 and MS12021082); in part by the Industry-University Cooperation Collaborative Education Projects under Grant 202102245013; and in part by the National College Students Innovation and Entrepreneurship Training Program under Grant No. 202110304042Z and Grant No. 202110304041Z.

**Institutional Review Board Statement:** Not applicable.

**Informed Consent Statement:** Not applicable.

**Data Availability Statement:** Not applicable.

**Acknowledgments:** The authors would like to thank the editor and the anonymous reviewers who provided insightful comments on improving this paper.

**Conflicts of Interest:** The authors declare no conflict of interest.

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
