# Peer review of "Structural Differences of PM2.5 Spatial Correlation Networks in Ten Metropolitan Areas of China"

_ijgi, doi:10.3390/ijgi11040267_

Round 1

Reviewer 1 Report

Since air pollution is cities is increasing with the intensity of human activities, this work does have merit to consider for publication however there are many issues that must be clarified and clarity must be improved. As it currently stands, novelty and international significance are missing due to the following reasons. Hence authors must revise the current version to demonstrate its international readership.

[1] Though abstract is well written, I would suggest using past tense to report your results in this section

[2] Introduction: This section is possibly unconvincing given the works you referred to show the motivation. Air pollution is not a problem in China only but also a severe issue in large cities especially in global south. However, you referred works from Chinese cities and thus cannot justify motivation is enhanced. I would suggest linking following works to enhance motivation and international significance of this work. It is also good to refer association or effect of air pollution on health, climate perturbations.  https://doi.org/10.1007/978-3-030-55092-9_13; https://doi.org/10.1016/j.pce.2022.103119; https://doi.org/10.1016/j.jastp.2018.01.022; https://doi.org/10.1016/j.dib.2019.105089; https://doi.org/10.1021/es400744g

[3] Data and methods: Would be useful if some description of air quality data is provided in section 2.2. Have you has a chance to look into this database https://wustl.app.box.com/v/ACAG-V5GL02-GWRPM25? Would be good to compare your findings with this. Line 145; instead of average ‘mean’ would be better to use. Sub-section 2.3.1 -2.3.2: I differ with you that you did not propose a new method rather you used existing methods in different context hence you must tweak your statements wherever appropriate in the texts. Subsection 2.3.4: If you control intercept and slope would be expect differing outcomes for factors affecting air pollution in the cities examined?

[4] Results: too long and easy to deviate from major points. Trim it and concentrate on useful outcomes that readers should know given there are hundred and thousands of air pollution related works in China

[5] Discussion: The poorest section indeed. In this section you did not cross check results of your work rather you provided two sections that are not that relevant. Using a few lines you can say these two items eg. suggestion and limitation. I would suggest to use above works and other similar works to demonstrate how your findings differ from existing works, and why there is difference (if any). Suggestion should be replaced with ‘Recommendation’ and should be ‘Limitations’ Don’t you think industrial activities in the cities are a major contributor to PM2.5 in the cities analysed?

Reviewer 2 Report

Dear authors,
The article presented for review takes up a very important and interesting topic and is a novel attempt to combine SNA with spatial analysis of PM2.5 pollutants. However, in its present form the article requires some additions and corrections in order to be published in a good scientific journal. I list the comments in points:
1) literature review - it should definitely go beyond the local context and refer to the rich world literature in this field relating both to the phenomenon itself (PM2.5 pollution), its factors as well as methods of study and analysis.
2) If factors are mentioned and then their selection for analysis, their selection should be justified. For example, it is not clear why the average temperature was used in the analysis and not, for example, some indicator related to precipitation or wind, which are more important factors for PM2.5 level and spread. One of the important pollution factors is mobility (transport) even if there is no separate indicator for it, it is worth indicating that GDP or population is an approximation.
3) The use of the SNA method should be justified and this should be done in the light of the literature, especially that it is not an obvious method for this type of analysis and this type of data.
4) From the description of the methods of spatial analysis it is not clear for how many points in each metropolitan area data were obtained and calculations were carried out. It is also unclear how data relating to areas (administrative units) were assigned to points for the calculation of gravity measures (centroids?, grids?). 

5) Figure 4 is unclear. Figure 3 should be divided into 3 separate Figures. Figure 2 should be larger to be readable.
6) No convincing discussion has been provided that aggregating data to whole year averages for PM2.5 makes sense. The risk is that they are highly variable over the year (depending on e.g. weather, seasonality of some economic activities etc.).
6) The conclusions in section 4 (Discussion) although generally correct do not follow directly from the analysis. 

To sum up - the conducted research is an interesting attempt, however, a rather selective treatment of factors and not much attention paid to the spatial aspect of the conducted research, as well as the influence of the applied method on the results, raise some doubts and require further corrections.

With best regards,

Reviewer 3 Report

The paper is well structured and it is about a new approach to evaluate air quality based on socio-economic factor in several Chinese regions over few years of observations. This new approach is very useful in order to understand how it is possible to tackle air quality issues considering the geography, demography and socio-economic scenario in a given area.

While the paper is very interesting and offer a challenging research, it lacks of few important and fundamental explanations that should be states in the abstract, in the introduction and also in the material and methods.

From the beginning of the text, the authors talks about a “network” that, later on, they say it has been represented in the form of a graph. So, as usually made with road network, it is important to state and clearly explain (from the beginning) what is this network about. Which are the nodes of this network? The cities?. Further, it should be very useful to show a graphical representation (on a map) of this network of cities. Therefore, it is important to show a graph with cities  as nodes and which kind of edges (links) have been chosen and with which criterium.

In the following the network analysis has been well explained with the betweenness centrality and the closeness centrality and its link with PM2.5 levels.

Therefore, considering the high originality of this work, I consider this paper as a valuable work to be published in International Journal of Geo-information but after minor correction/implementation as stated above.

Round 2

Reviewer 1 Report

Thank you 

Author Response

Thank you for your review of this article, we have made a detailed revision according to the academic editor's opinions.

Reviewer 2 Report

Dear authors,
Thank you for your answers and explanations. The revised version of the article is much better and should be published.
Best regards

Author Response

(The authors gave the same response as above.)
